# Minimizers of the Empirical Risk
# and Risk Monotonicity

**Marco Loog**
Delft University of Technology & University of Copenhagen

**Tom Viering**
Delft University of Technology

**Alexander Mey**
Delft University of Technology

## Abstract

Plotting a learner's average performance against the number of training samples results in a learning curve. Studying such curves on one or more data sets is a way to get to a better understanding of the generalization properties of this learner. The behavior of learning curves is, however, not very well understood and can display (for most researchers) quite unexpected behavior. Our work introduces the formal notion of *risk monotonicity*, which asks the risk to not deteriorate with increasing training set sizes in expectation over the training samples. We then present the surprising result that various standard learners, specifically those that minimize the empirical risk, can act *non*monotonically irrespective of the training sample size. We provide a theoretical underpinning for specific instantiations from classification, regression, and density estimation. Altogether, the proposed monotonicity notion opens up a whole new direction of research.

## 1  Introduction

Learning curves are an important diagnostic tool that provide researchers and practitioners with insight into a learner's generalization behavior [Shalev-Shwartz and Ben-David, 2014]. Learning curves plot the (estimated) true performance against the number of training samples. Among other things, they can be used to compare different learners to each other. This can highlight the differences due to their complexity, with the simpler learners performing better in the small sample regime, while the more complex learners perform best with large sample sizes. In combination with a plot of their (averaged) resubstitution error (or training error), they can also be employed to diagnose underfitting and overfitting. Moreover, they can aid when it comes to making decision about collecting more data or not by extrapolating them to sample sizes beyond the ones available.

It seems intuitive that learners become better (or at least do not deteriorate) with more training data. With a bit more reservation, Shalev-Shwartz and Ben-David [2014] state, for instance, that the learning curve "must start decreasing once the training set size is larger than the VC-dimension" (page 153). The large majority of researchers and practitioners (that we talked to) indeed take it for granted that learning curves show improved performance with more data. Any deviations from this they contribute to the way the experiments are set up, to the finite sample sizes one is dealing with, or to the limited number of cross-validation or bootstrap repetitions one carried out. It is expected that if one could sample a training set *ad libitum* and measure the learner's *true* performance over all data, such behavior disappears. That is, if one could indeed get to the performance in expectation over all test data and over all training samples of a particular size, performance supposedly improves with more data.

We formalize this behavior of expected improved performance in Section 3. As we will typically express a learner's efficiency in term of the expected loss, we will refer to this notation as *risk monotonicity*. Section 4 then continues with the main contribution of this work and demonstrates that various well-known empirical risk minimizers can display nonmonotonic behavior. Moreover, we show that for these learners this behavior can persist indefinitely, i.e., it can occur at any sample size. *Note*: all proofs can be found in the supplement. Section 5 provides some experimental evidence for some cases of interest that have, up to now, resisted any deeper theoretical analysis. Section 6 then provides a discussion and concludes the work. In this last section, among others, we contrast our notion of risk monotonicity to that of PAC-learnability, note that these are two essentially different concepts, and consider various research questions of interest to further refine our understanding of learning curves. Though many will probably find our findings surprising, counterintuitive behavior of the learning curve has been reported before in various other settings. Section 2 goes through these and other related works and puts our contribution in perspective.

## 2   Earlier Work and Its Relation to the Current

We split up our overview into the more regular works that characterize monotonic behavior and those that identify the existence of nonmonotonic behavior.

### 2.1   The Monotonic Character of Learning Curves

Many of the studies into the behavior of learning curves stem from the end of the 1980s and the beginning of the 1990s and were carried out by Tishby, Haussler, and others [Tishby et al., 1989, Levin et al., 1990, Sompolinsky et al., 1990, Opper and Haussler, 1991, Seung et al., 1992, Haussler et al., 1992]. These early investigations were done in the context of neural networks and in their analyses typically make use of tools from statistical mechanics. A statistical inference approach is studied by Amari et al. [1992] and Amari and Murata [1993], who demonstrate the typical power-law behavior of the asymptotic learning curve. Haussler et al. [1996] bring together many of the techniques and results from the aforementioned works. At the same time, they advance the theory for learning curves and provide an overview of the rather diverse, though still monotonic, behavior they can exhibit. In particular, the curve may display multiple steep and sudden drops in the risk.

Already in 1979, Micchelli and Wahba [1979] provide a lower bound for learning curves of Gaussian processes. Only at the end of the 1990s and beginning of the 2000s, the overall attention shifted from neural networks to Gaussian processes. In this period, various works were published that introduce approximations and bounds [Opper, 1998, Sollich, 1999, Opper and Vivarelli, 1999, Williams and Vivarelli, 2000, Sollich and Halees, 2002]. Different types of techniques were employed in these analyses, some of which again from statistical mechanics. The main caveat, when it comes to the results obtained, is the assumption that the model is correctly specified.

The focus of Cortes et al. [1994] is on support vector machines. They develop efficient procedures for an extrapolation of the learning curve, so that if only limited computational resources are available, these can possibly be assigned to the most promising approaches. It is assumed that, for large enough training set sizes, the error rate converges towards a stable value following a power-law. This behavior was established to hold in many of the aforementioned works. The ideas that Cortes et al. [1994] put forward have found use in specific applications (see, for instance, [Kolachina et al., 2012]) and can count on renewed interest these days, especially in combination with flop gobbling neural networks (see, for instance, [Hestness et al., 2017]).

All of the aforementioned works study and derive learning curve behavior that shows no deterioration with growing training set sizes, even though they may be described as "learning curves with rather curious and dramatic behavior" [Haussler et al., 1996]. Our work identifies aspects that are more curious and more dramatic: with a larger training set, performance can deteriorate, even in expectation.

### 2.2   Early Noted Nonmonotonic Behavior

Probably the first to point out that learning curves can show nonmonotonic behavior was Duin [1995], who looked at the error rate of so-called Fisher's linear discriminant. In this context, Fisher's linear discriminant is used as a classifier and equivalent to the two-class linear classifier that is obtained by optimizing the squared loss. This can be solved by regressing the input feature vectors onto

a $-1/+1$ encoding of the class labels. In case the number of training samples is smaller than or equal to the number of input dimensions, one needs to deal with the inverse of singular matrices and typically resorts to the use of the Moore-Penrose pseudo-inverse. In this way, the minimum norm solution is obtained [Smola et al., 2000]. It is exactly in this underdetermined setting, as the number of training samples approaches the dimensionality, that the error rate will be increasing. Around the same time, Opper and Kinzel [1996] showed that in the context of neural networks a similar behavior is observed for small samples. In particular, the error rate for the single layer perceptron is demonstrated to increase when the training set size goes towards the dimensionality of the data [Opper, 2001]. Subsequently, other examples of exactly this type of nonmonotonic behavior have been reported. Worth mentioning are classifiers built based on the lasso [Krämer, 2009] and two recent works that have trigger renewed attention to this subject in the neural networks community [Belkin et al., 2018, Spigler et al., 2018]. The classifier reaching a maximum error rate when the sample size transits from an underspecified to an overspecified setting is originally referred to as peaking (see also [Duin, 2000]). The two recent works above rename it and use the terms double descent and jamming.

A completely different phenomenon, and yet other way in which learning curves can be nonmonotonic, is described by Loog and Duin [2012]. They show that there are learning problems for which specific classifiers attain their optimal expected 0-1 loss at a finite sample size. That is, on such problems, these classifiers perform essentially worse with an infinite amount of training data compared to some finite training set sizes. The behavior is referred to as dipping, following the shape of the error rate's learning curve. In the context of (safe) semi-supervised learning, Loog [2016] then argues that if one cannot even guarantee improvements in 0-1 loss when receiving more labeled data, this is certainly impossible with unlabeled data. When evaluating in terms of the loss the model optimizes, however, one can get to demonstrable improvements and essentially solve the safe semi-supervised leaning problem [Loog, 2016, Krijthe and Loog, 2017, 2018]. Our work shows, however, that also when one looks at the loss the learner optimizes, there may be no performance guarantees.

The dipping behavior hinges both on the fact that the model is misspecified (i.e., the Bayes-optimal estimate is not in the class of models considered) and that the classifier does not optimize what it is ultimately evaluated with. That this setting can cause problems, e.g. convergence to the wrong solution, had already been demonstrated for maximum likelihood by Devroye et al. [1996]. If the model class is flexible enough, this discrepancy disappears in many a setting. This happens, for instance, for the class of classification-calibrated surrogate losses [Bartlett et al., 2006]. Note, however, that Devroye et al. [1996] conjecture that consistent rules that are expected to perform better with increasing training sizes (so-called smart rules) do not exist. Ben-David et al. [2012] analyze the consequence of the mismatch between surrogate and zero-one loss in some more detail and provide another example of a problem distribution on which such classifiers would dip.

Our results strengthen or extend the above findings in the following ways. First of all, we show that nonmonotonic behavior can occur in the setting where the complexity of the learner is small compared to the training set size. Therefore, the reported behavior is not due to jamming or peaking. Secondly, we are going to evaluate our learners by means of the loss they actually optimize for. If we look at the linear classifier that optimizes the hinge loss, for instance, we will study its learning curve for the hinge loss as well. In other words, there is no discrepancy between the objective used during training and the loss used at test time. Therefore, possibly odd behavior cannot be explained by dipping. As a third, we do not only look at classification and regression but also consider density estimation and (negative) log-likelihood estimation in particular.

## 3 Risk Monotonicity

We come to a formal definition of the intuition that with one additional instance a learner should improve its performance in expectation over the training set. The next section then study various learners with the notions developed here. First, however, some notations and prior definitions are provided.

### 3.1 Preliminaries

We let $S_n = (z_1, \ldots, z_n)$ be a training set of size $n$, sampled i.i.d. from a distribution $D$ over a general domain $\mathscr{Z}$. Also given is a hypothesis class $\mathscr{H}$ and a loss function $\ell : \mathscr{Z} \times \mathscr{H} \to \mathbb{R}$ through which

the performance of a hypothesis $h \in \mathcal{H}$ is measured. The objective is to minimize the expected loss or risk under the distribution $D$, which is given by

$$R_D(h) := \mathop{\mathbb{E}}_{z \sim D} \ell(z, h). \tag{1}$$

A learner $A$ is a particular mapping from the set of all samples $\mathcal{S} := \mathcal{Z} \cup \mathcal{Z}^2 \cup \mathcal{Z}^3 \cup \ldots$ to elements from the prespecified hypothesis class $\mathcal{H}$. That is, $A : \mathcal{S} \to \mathcal{H}$. We are particularly interested in learners $A_{\mathrm{erm}}$ that provide a solution which minimizes the empirical risk $R_{S_n}$ over the training set:

$$A_{\mathrm{erm}}(S_n) := \operatorname*{argmin}_{h \in \mathcal{H}} R_{S_n}(h), \tag{2}$$

with

$$R_{S_n}(h) := \frac{1}{n} \sum_{i=1}^{n} \ell(z_i, h). \tag{3}$$

Most common classification, regression, and density estimation problems can be formulated in such terms. Examples are the earlier mentioned Fisher's linear discriminant, support vector machines, and Gaussian processes, but also maximum likelihood estimation, linear regression, and the lasso can be cast in similar terms.

## 3.2 Degrees of Monotonicity

The basic definition is the following.

**Definition 1 (local monotonicity)** *A learner A is $(D, \ell, n)$-monotonic with respect to a distribution $D$, a loss $\ell$, and an integer $n \in \mathbb{N} := \{1, 2, \ldots\}$ if*

$$\mathop{\mathbb{E}}_{S_{n+1} \sim D^{n+1}} [R_D(A(S_{n+1})) - R_D(A(S_n))] \leq 0. \tag{4}$$

This expresses exactly how we would expect a learner to behave locally (i.e., at a specific training sample size $n$): given one additional training instance, we expect the learner to improve. Based on our definition of local monotonicity, we can construct stronger desiderata that may be of more interest.

The two entities we would like to get rid of in the above definition are $n$ and $D$. The former, because we would like our learner to act monotonically irrespective of the sample size. The latter, because we typically do not know the underlying distribution. For now, getting rid of the loss $\ell$ is maybe too much to ask for. First of all, not all losses are compatible with one another, as they may act on different types of $z \in \mathcal{Z}$ and $h \in \mathcal{H}$. But even if they take the same types of input, a learner is typically designed to minimize one specific loss and there seems to be no direct reason for it to be monotonic in terms of another. It seems less likely, for example, that an SVM is risk monotonic in terms of the squared loss. (We will nevertheless briefly return to this matter in Section 6.) We exactly focus on the empirical risk minimizers as they seem to be the most appropriate candidates to behave monotonically in terms of their own loss.

Though we typically do not know $D$, we do know in which domain $\mathcal{Z}$ we are operating. Therefore, the following definition is suitable.

**Definition 2 (local $\mathcal{Z}$-monotonicity)** *A learner A is (locally) $(\mathcal{Z}, \ell, n)$-monotonic with respect to a loss $\ell$ and an integer $n \in \mathbb{N}$ if, for all distributions $D$ on $\mathcal{Z}$, it is $(D, \ell, n)$-monotonic.*

When it comes to $n$, the peaking phenomenon shows that, for some learners, it may be hopeless to demand local monotonicity for all $n \in \mathbb{N}$. What we still can hope to find is an $N \in \mathbb{N}$, such that for all $n \geq N$, we find the learner to be locally risk monotonic. As properties like peaking may change with the dimensionality—the complexity of the classifier is generally dependent on it, the choice for $N$ will typically have to depend on the domain.

**Definition 3 (weak $\mathcal{Z}$-monotonicity)** *A learner A is weakly $(\mathcal{Z}, \ell, N)$-monotonic with respect to a loss $\ell$ if there is an integer $N \in \mathbb{N}$ such that for all $n \geq N$, the learner is locally $(\mathcal{Z}, \ell, n)$-monotonic.*

Given the domain, one may of course be interested in the smallest $N$ for which weak $\mathcal{Z}$-monotonicity is achieved. If it does turn out that $N$ can be set to 1, the learner is said to be globally $\mathcal{Z}$-monotonic.

**Definition 4 (global $\mathcal{Z}$-monotonicity)** *A learner A is globally $(\mathcal{Z}, \ell)$-monotonic with respect to a loss $\ell$ if for every integer $n \in \mathbb{N}$, the learner is locally $(\mathcal{Z}, \ell, n)$-monotonic.*

# 4 Theoretical Results

We consider the hinge loss, the squared loss, and the absolute loss and linear models that optimize the corresponding empirical loss. In essence, we demonstrate that, there are various domains $\mathscr{Z}$ for which for any choice of $N$, these learners are *not* weakly $(\mathscr{Z}, \ell, N)$-monotonic. For the log-likelihood, we basically prove the same: there are standard learners for which the (negative) log-likelihood is not weakly $(\mathscr{Z}, \ell, N)$-monotonic for any $N$. The first three losses can all be used to build classifiers: the first is at the basis of SVMs, while the second gives rise to Fisher's linear discriminant in combination with linear hypothesis classes. The second and third loss are of course also employed in regression. The log-likelihood is standard in density estimation.

## 4.1 Learners that Do Behave Monotonically

Before we actually move to our negative results, we first provide examples that point in a positive direction. The first learner is provably risk monotonic over a large collection of domains. The second learner, the memorize algorithm, is a monotonic learner taken from [Ben-David et al., 2011].

**Fitting a normal distribution with fixed covariance and unknown mean.** Let $\Sigma$ be an invertible $d \times d$-matrix,

$$\mathscr{H} := \left\{ z \mapsto \frac{1}{\sqrt{(2\pi)^d |\Sigma|}} \exp(-\tfrac{1}{2}(z-\mu)^T \Sigma^{-1}(z-\mu)) \,\middle|\, \mu \in \mathbb{R}^d \right\}, \tag{5}$$

$\mathscr{Z} \subset \mathbb{R}^d$, and take the loss to equal the negative log-likelihood.

**Theorem 1** *If $\mathscr{Z}$ is bounded, the learner $A_{\mathrm{erm}}$ is globally $(\mathscr{Z}, \ell)$-monotonic.*

**Remark 1** *Using similar arguments, one can show that the learner with $\mathscr{H} = \mathbb{R}^d$ and Mahalanobis loss $\ell(z,h) = ||z-h||_\Sigma^2 := (z-h)^T \Sigma(z-h)$, with $\Sigma$ a positive semi-definite matrix, is globally $(\mathscr{Z}, \ell)$-monotonic as well as long as $\mathscr{Z}$ is bounded.*

**The memorize algorithm [Ben-David et al., 2011].** When evaluated on a test input object that is also present in the training set, this classifier returns the label of said training object. In case multiple training examples share the same input, the majority voted label is returned. In case the test object is not present in the training set, a default label is returned. This learner is monotonic for any distribution under the zero-one loss. Similairly, any histogram rule with fixed partitions is monotone, which is immediate from the properties of the binomial distribution [Devroye et al., 1996].

## 4.2 Learners that Don't Behave

To show for various learners that they do not always behave risk monotonically, we construct specific discrete distributions for which we can explicitly proof nonmonotonicity. What leads to the sought-after counterexamples in our case, is a distribution where a small fraction of the density is located relatively far away from the origin. In particular, shrinking the probability of this fraction towards 0 leads us to the lemma below. It is used in the subsequent proofs, but is also of some interest in itself.

**Lemma 1** *Let $\mathscr{Z} := \{a,b\}$ be a domain with two elements from $\mathbb{R}$, let*

$$S_{n-k}^k := (\underbrace{a,\ldots,a}_{k \text{ elements}}, \underbrace{b,\ldots,b}_{n-k \text{ elements}}) \tag{6}$$

*be a training set with n samples, and let $h_{n-k}^k := A_{\mathrm{erm}}(S_{n-k}^k)$. If*

$$-\ell(b, h_{n+1}^0) + (n+1)\ell(b, h_n^1) - n\ell(b, h_{n-1}^1) > 0, \tag{7}$$

*then $A_{\mathrm{erm}}$ is not locally $(\mathscr{Z}, \ell, n)$-monotonic.*

**Remark 2** *For many losses, we have, in fact, that $\ell(b, h_n^0) = \ell(b, h_{n+1}^0) = 0$, which further simplifies the difference of interest to $(n+1)\ell(b, h_n^1) - n\ell(b, h_{n-1}^1)$.*

In a way, the above lemma and remark show that if the learning of the single point $b$ does not happen fast enough, local monotonicity cannot be guaranteed. Section 6 will briefly return to this point.

**Linear hypotheses, squared loss, absolute loss, and hinge loss.** We consider linear models without bias in $d$ dimensions, so take $\mathscr{Z} = \mathscr{X} \times \mathscr{Y} \subset \mathbb{R}^d \times \mathbb{R}$ and $\mathscr{H} = \mathbb{R}^d$. Though not crucial to our argument, we select the minimum-norm solution in the underdetermined case. $A_{\mathrm{erm}} : \mathscr{H} \to \mathbb{R}^d$ is the general minimizer of the risk in this setting. For the squared loss, we have $\ell(z,h) = (x^T h - y)^2$ for any $z = (x,y) \in \mathscr{Z}$. The absolute loss is given by $\ell(z,h) = |x^T h - y|$ and the hinge loss is defined as $\ell(z,h) = \max(0, 1 - yx^T h)$. Both the absolute loss and the squared loss can be used for regression and classification. The hinge loss is appropriate only for the classification setting. Therefore, though the rest of the setup remains the same, outputs are limited to the set $\mathscr{Y} = \{-1, +1\}$ for the hinge loss.

**Theorem 2** *Consider a linear $A_{\mathrm{erm}}$ without intercept and assume it either optimizes the squared, the absolute, or the hinge loss. Assume $\mathscr{Y}$ contains at least one nonzero element. If there exists an open ball $B_0$ that contains the origin, such that $B_0 \subset \mathscr{X}$, then this risk minimizer is* not *weakly $(\mathscr{Z}, \ell, N)$-monotonic for any $N \in \mathbb{N}$.*

**Fitting a normal distribution with fixed mean and unknown variance (in one dimension).** We follow up on the example where we fitted a normal distribution with fixed covariance and unknown mean. We limit ourselves, however, to one dimension only and, more importantly, now take the variance to be the unknown, while fixing the mean (to 0, arbitrarily). Specifically, let $\mathscr{H} := \{z \mapsto \frac{1}{\sqrt{2\pi\sigma^2}} \exp(-\frac{1}{2\sigma^2}z^2) | \sigma > 0\}$, $\mathscr{Z} \subset \mathbb{R}$, and take the loss to equal the negative log-likelihood.

**Theorem 3** *If there exists an open ball $B_0$ that contains the origin, such that $B_0 \subset \mathscr{Z}$, then estimating the variance of a one-dimensional normal density is* not *weakly $(\mathscr{Z}, \ell, N)$-monotonic for any $N \in \mathbb{N}$.*

## 5    Experimental Evidence

Our results from the previous section, already show cogently that the behavior of the learning curve can be interesting to study. Here we complement our theoretical findings with a few illustrative experiments to strengthen this point even further. The results can be found in Figure 1, which displays (numerically) exact learning curves for a couple of different settings.

The input space considered for all our examples is one-dimensional. The experiment in Subfigure 1b relies on the absolute loss, while all other make use of the squared loss. In addition, Subfigures 1a, 1b, and 1c consider distributions with two points: $a = (1, 1)$ and $b = (\frac{1}{10}, 1)$ with the first coordinate the input and the second the corresponding output. Different plots use different values for the probability of observing $a$. For Subfigure 1a, $P(a) = 0.00001$, Subfigure 1b uses $P(a) = 0.1$, and Subfigure 1c takes $P(a) = 0.01$. For Subfigure 1c, we also studied the effect of a small amount of standard $L_2$-regularization decreasing with training size ($\lambda = \frac{0.01}{n}$), leading to the regularized solution $A_{\mathrm{reg}}$. The distribution for Subfigure 1d is slightly different and supported on three points: $a = (1, 1)$, $b = (\frac{1}{10}, -1)$, and $c = (-1, 1)$, with again the first coordinate as the input and the second the corresponding output. In this case, $P(a) = 0.01$, $P(b) = 0.01$, and $P(c) = 0.98$. This last experiment concerns least squares regression with a bias term: a setting we have not been able to analyze theoretically up to this point.

Most salient is probably the serrated and completely nonmonotonic behavior of the learning curve for the absolute loss in Figure 1b. Of interest as well is that regularization does not necessarily solve the problem. Subfigure 1c even shows it can make it worse: $A_{\mathrm{reg}}$ gives nonmonotonic behavior, while $A_{\mathrm{erm}}$ is monotonic under the same distribution (cf. [Grünwald and Kotłowski, 2011]). Subfigure 1a illustrates clearly how dramatic the expected squared loss can grow with more data.

In the final example in Figure 1d, as already noted, we consider linear regression with the squared loss that includes a bias term in combination with the distribution supported on three points. This example is of interest because the usual configuration for standard learners includes such bias term and one could get the impression from our theoretical results (and maybe in particular the proofs) that the origin plays a major role in the bad behavior of some of the learners. But as can be observed here, adding an intercept, and therefore taking away the possibly special status of the origin does not make risk nonmonotonicity go away.

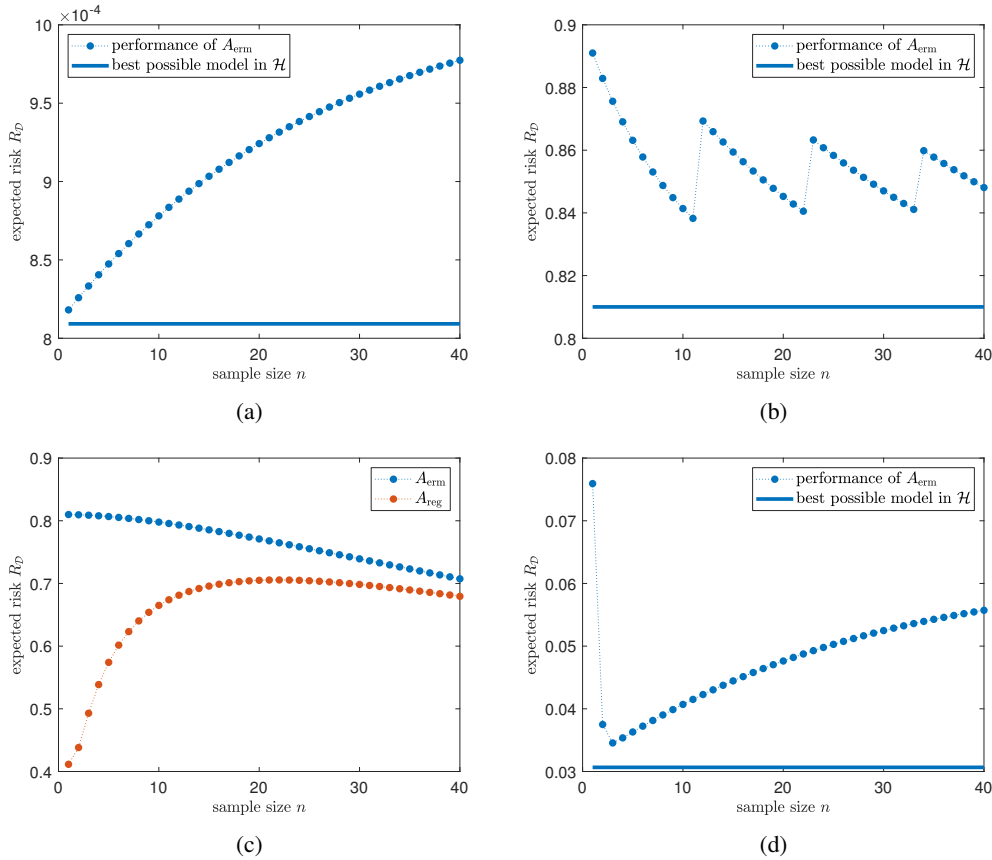

Figure 1: Learning curves (average risk against training set size) for some one-dimensional problems. Subfigure (a) is based on squared loss, no intercept; (b) on absolute loss, no intercept; (c) on squared loss, no intercept (with and without regularization); (d) on squared loss with intercept. The dashed line, indicates the risk the learner attains in the limit of an infinite training set size.

## 6 Discussion and Conclusion

It should be clear that this paper does not get to the bottom of the learning-curve issue. In fact, one of the reasons of this work is to bring it to the attention of the community. We are convinced that it raises a lot of interesting and interrelated problems that may go far beyond the initial analyses we offer here. Further study should bring us to a better understanding of how learning curves can actually act, which, in turn, should enable practitioners to better interpret and anticipate their behavior.

What this work does convey is that learning curves can (provably) show some rather counterintuitive and surprising behavior. In particular, we have demonstrated that least squares regression, regression with the absolute loss, linear models trained with the hinge loss, and likelihood estimation of the variance of a normal distribution can all suffer from nonmonotonic behavior, even when evaluated with the loss they optimize for. All of these are standard learners, using standard loss functions.

Anyone familiar with the theory of PAC learning may wonder how our results can be reconciliated with the bounds that come from this theory. At a first glance, our observations may seem to contradict this theory. Learning theory dictates that if the hypothesis class has finite VC-dimension, the excess risk $\varepsilon$ of ERM will drop as $\varepsilon = O(\frac{1}{n})$ in the realizable case and as $\varepsilon = O(\frac{1}{\sqrt{n}})$ in the agnostic case [Vapnik, 1998, Shalev-Shwartz and Ben-David, 2014]. Thus PAC bounds give an upper bound on the excess risk $\varepsilon$ that will be tighter given more samples. PAC bounds hold with a particular probability, but we are concerned with the risk in expectation. Even bounds that hold in expectation over the training sample will, however, not rule out nonmonotonic behavior. This is because in the end the guarantees from PAC learning are indeed merely bounds. Our analysis show that within those

bounds, we cannot always expect risk monotonic behavior. In fact, learning problems of all four possible combinations exist: not PAC-learnable and monotonic, PAC-learnable and not monotonic, etc. For instance, the memorize algorithm (end of Subsection 4.1) is monotone, while it has infinite VC-dimension and so is not PAC-learnable.

In light of the learning rates mentioned above, we wonder whether there are deeper links with Lemma 1 (see also Remark 2). Rewrite Equation (7) to find that we do not have local monotonicity at $n$ in case

$$\frac{-\frac{\ell(b,h_{n+1}^0)}{n+1} + \ell(b,h_n^1)}{\ell(b,h_{n-1}^1)} > \frac{n}{n+1}. \tag{8}$$

With $n$ large enough, we can ignore the first term in the numerator. So if a learner, in this particular setting, does not learn an instance $b$ at least at a rate of $\frac{n}{n+1}$ in terms of the loss, it will display nonmonotonic behavior. According to learning theory, for agnostic learners, the fraction between two subsequent losses is of the order $\sqrt{\frac{n}{n+1}}$, which is always larger than $\frac{n}{n+1}$ for $n > 0$. Can one therefore generally expect nonmonotonic behavior for any agnostic learner? Our normal mean estimation problem shows it cannot. But then, what is the link, if any?

As already hinted at in the introduction, our findings may also warrant revisiting the results obtained in [Loog, 2016, Krijthe and Loog, 2017, 2018]. These works show that there are some semi-supervised learners that allow for essentially improved performance over the supervised learner, i.e., these are truly safe. Though this is the transductive setting, this may in a sense just shows how strong these results are. In the end, their estimation procedures is really rather different from empirical risk minimization, but it does beg the question whether similar constructs can be used to get to risk monotonic procedures in the supervised case.

Another question, related to the last remark above, seems of interest: could it be that the use of particular losses at training time leads to monotonic behavior at test time? Or can regularization still lead to more monotonic behavior, e.g. by explicitly limiting $\mathcal{H}$? Maybe particular (upper-bounding) convex losses could turn out to behave risk monotonic in terms of specific nonconvex losses? Dipping seems to show, however, that this may very well not be the case. Results concerning smart rules, i.e., classifiers that act monotonically in terms of the error rate [Devroye et al., 1996], seem to point in the same direction. So should we expect it to be the other way round? Can nonconvex losses bring us monotonicity guarantees for convex ones? Of course, monotonicity properties of *nonconvex* learners are also of interest to study in their own respect.

An ultimate goal would of course be to fully characterize when one can have risk monotonic behavior and when not. At this point we do not have a clear idea to what extent this would at all be possible. We were, for instance, not able to analyze some standard, seemingly simple cases, e.g. simultaneously estimating the mean and the variance of a normal model. And maybe we can only get to rather weak results. Only knowledge about the domain may turn out to be insufficient and we need to make assumptions on the class of distributions $\mathcal{D}$ we are dealing with (leading to some notion of weakly $\mathcal{D}$-monotonicity?). For a start, we could study likelihood estimation under correctly specified models, for which generally there turn out to be remarkably few finite-sample results. One can also wonder whether it is possible to find salient distributional properties that can be specifically related to the overall shape of the learning curve (see, for instance, [Haussler et al., 1996]).

All in all, we believe that our theoretical results, strengthened by some illustrative examples, show that the monotonicity of learning curves is an interesting and nontrivial property to study.

**Acknowledgments**

We received various suggestions and comments, among others based on an abstract presented earlier [Viering et al., 2019]. We particularly want to thank Peter Grünwald, Steve Hanneke, Wojciech Kotłowski, Jesse Krijthe, and David Tax for constructive feedback and discussions.

This work was funded in part by the Netherlands Organisation for Scientific Research (NWO) and carried out under TOP grant project number 612.001.402.

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
