[Supplementary Material]

## Supplement: Proofs

### Theorem 1

The minimizing hypothesis of the empirical risk $A_{\text{erm}}(S_n)$ is attained for the mean that equals $\mu_n := \frac{1}{n}\sum_{i=1}^{n} z_i$. Equivalently, we have $\mu_{n+1} := \frac{1}{n+1}\sum_{i=1}^{n+1} z_i$ for the parameter value that defines $A_{\text{erm}}(S_{n+1})$. Let $F$ be the true cumulative distribution function for a single observation $z$ and let $F_n$ be the true cumulative distribution function for $\mu_n$. For simplicity, in what follows, all integrals are taken over $\mathbb{R}^d$ and the density outside of $\mathscr{Z}$ is simply taken to be equal to 0. The negative log-likelihoods, in expectation over the samples $S_n$, equals

$$-\int\int \log\left(\frac{1}{\sqrt{(2\pi)^d|\Sigma|}}\exp(-\tfrac{1}{2}(z-\mu_n)^T\Sigma^{-1}(z-\mu_n))\right)dF(z)dF_n(\mu_n)=$$

$$-\log\left(\frac{1}{\sqrt{(2\pi)^d|\Sigma|}}\right)+\int \tfrac{1}{2}z^T\Sigma^{-1}zdF(z)-\int\int z^T\Sigma^{-1}\mu_n dF(z)dF_n(\mu_n)+\int \tfrac{1}{2}\mu_n^T\Sigma^{-1}\mu_n dF_n(\mu_n)=$$

$$-\log\left(\frac{1}{\sqrt{(2\pi)^d|\Sigma|}}\right)+\int \tfrac{1}{2}z^T\Sigma^{-1}zdF(z)-\mu^T\Sigma^{-1}\mu+\int \tfrac{1}{2}\mu_n^T\Sigma^{-1}\mu_n dF_n(\mu_n)$$

Following Equation (4), we consider the difference between the above term and the one corresponding to $n+1$ training samples. As only the last term differs in the expressions for $n$ and $n+1$ samples, we find that this difference equals

$$\int \tfrac{1}{2}\mu_{n+1}^T\Sigma^{-1}\mu_{n+1}dF_{n+1}(\mu_{n+1})-\int \tfrac{1}{2}\mu_n^T\Sigma^{-1}\mu_n dF_n(\mu_n). \tag{9}$$

$\mathscr{Z}$ is bounded, so the (noncentral) second moment matrix $M$ exists and the difference simplifies to

$$\int \tfrac{1}{2}\text{tr}\left(\mu_{n+1}\mu_{n+1}^T\Sigma^{-1}\right)dF_{n+1}(\mu_{n+1})-\int \tfrac{1}{2}\text{tr}\left(\mu_n\mu_n^T\Sigma^{-1}\right)dF_n(\mu_n)= \tag{10}$$

$$\int \tfrac{1}{2}\text{tr}\left(\frac{1}{(n+1)^2}\sum_{i=1}^{n+1}z_i z_i^T\Sigma^{-1}\right)dF_{n+1}(\mu_{n+1})-\int \tfrac{1}{2}\text{tr}\left(\frac{1}{n^2}\sum_{i=1}^{n}z_i z_i^T\Sigma^{-1}\right)dF_n(\mu_n)= \tag{11}$$

$$\tfrac{1}{2}\text{tr}\left(\frac{1}{(n+1)}M\Sigma^{-1}\right)-\tfrac{1}{2}\text{tr}\left(\tfrac{1}{n}M\Sigma^{-1}\right)\leq 0. \tag{12}$$

This proves that the learner is globally $(\mathscr{Z},\ell)$-monotonic. $\square$

### Lemma 1

Let $P(a)=q$ and $P(b)=1-q$. The expected risk over $S_n$ then equals

$$R(q):=\sum_{k=0}^{n}\binom{n}{k}q^k(1-q)^{n-k}\left(q\ell(a,h_{n-k}^k)+(1-q)\ell(b,h_{n-k}^k)\right). \tag{13}$$

The derivative to $q$ of the above equals

$$\frac{d}{dq}R(q)=\sum_{k=0}^{n}\binom{n}{k}\Big[(k+1)q^k(1-q)^{n-k}\ell(a,h_{n-k}^k)-(n-k)q^{k+1}(1-q)^{n-k-1}\ell(a,h_{n-k}^k)+$$

$$kq^{k-1}(1-q)^{n-k+1}\ell(b,h_{n-k}^k)-(n-k+1)q^k(1-q)^{n-k}\ell(b,h_{n-k}^k)\Big]. \tag{14}$$

Taking the limit $q\to 0$, all terms become zero for $k>1$. For $k=0$, we get $\ell(a,h_n^0)-(n+1)\ell(b,h_n^0)$ and, for $k=1$, we get $n\ell(b,h_{n-1}^1)$. Similarly, for a training sample size of $n+1$, the only nonzero terms we get are for $k\in\{0,1\}$, as the expression for the derivative is essentially the same.

It shows that the $q$-derivative evaluated in 0 of the difference in expected risk from Equation (4) equals $\ell(a,h_{n+1}^0)-(n+2)\ell(b,h_{n+1}^0)+(n+1)\ell(b,h_n^1)-\ell(a,h_n^0)+(n+1)\ell(b,h_n^0)-n\ell(b,h_{n-1}^1)$, which can be further simplified to $-\ell(b,h_{n+1}^0)+(n+1)\ell(b,h_n^1)-n\ell(b,h_{n-1}^1)$, as $\ell(a,h_n^0)=\ell(a,h_{n+1}^0)$ and $\ell(b,h_n^0)=\ell(b,h_{n+1}^0)$.

If this derivative is strictly larger than 0, continuity in $q$ implies that there is a $q>0$ such that the actual risk difference becomes positive. This shows that $A_{\text{erm}}$ is not locally $(\mathscr{Z},\ell,n)$-monotonic. $\square$

**Theorem 2**

Let us first consider the squared loss. Take $a = (a_1, 0, \ldots, 0, a_{d+1})$ and $b = (b_1, 0, \ldots, 0, b_{d+1})$, such that the input vectors, $(a_1, 0, \ldots, 0)$ and $(b_1, 0, \ldots, 0)$, which constitute the first $d$ coordinates are in $B_0 \subset \mathscr{Z}$. The variables $a_{d+1}$ and $b_{d+1}$ constitute the outputs. Let both first input coordinates $a_1$ and $b_1$ not be equal to 0. All other input coordinates do equal 0. In this case, all (minimum-norm) hypotheses are finite and Remark 2 applies to this setting. So we study whether $(n+1)\ell(b, h_n^1) - n\ell(b, h_{n-1}^1) > 0$ in order to be able to invoke Lemma 1. To do so, we exploit that we can determine $h_n^1$ in closed-form. As all input variation occurs in the first coordinate only, we have that $h_n^1 = \left( \frac{a_1 a_{d+1} - n b_1 b_{d+1}}{a_1^2 + n b_1^2}, 0, \ldots, 0 \right) \in \mathbb{R}^d$, which implies that $\ell(b, h_n^1) = \left( b_1 \frac{a_1 a_{d+1} - n b_1 b_{d+1}}{a_1^2 + n b_1^2} - b_{d+1} \right)^2$. In the same way we, find that $\ell(b, h_{n-1}^1) = \left( b_1 \frac{a_1 a_{d+1} - (n-1) b_1 b_{d+1}}{a_1^2 + (n-1) b_1^2} - b_{d+1} \right)^2$. Now take the limit of $b_1$ to 0 to obtain $(n+1)\ell(b, h_n^1) - n\ell(b, h_{n-1}^1) = (n+1)b_{d+1}^2 - n b_{d+1}^2 = b_{d+1}^2$. For any $b_{d+1}$ bounded away from 0, this shows that for all $n \in \mathbb{N}$ there is a $b_1 > 0$, small enough, such that $(n+1)\ell(b, h_n^1) - n\ell(b, h_{n-1}^1) > 0$. This shows in turn that there exists a $b_1$ and a corresponding $b_{d+1} \neq 0$, such that $A_{\text{erm}}$ under the squared loss is not locally $(\mathscr{Z}, \ell, n)$-monotonic. As this holds for all $n$, we conclude that it also is not weakly $(\mathscr{Z}, \ell, N)$-monotonic for any $N \in \mathbb{N}$.

For the absolute loss, we consider the same setting as for the squared loss and its very beginning proceeds along the exact same lines. The proof starts to deviate at the calculation of $\ell(b, h_n^1)$ and $\ell(b, h_{n-1}^1)$. Still the same as for the squared loss, as all input variation occurs in the first coordinate, we only have to study what happens in that subspace. This means that all other $d-1$ elements of the minimum-norm solutions we consider will be 0. As $h_n^1$ is the empirical risk minimizer for one $a$ and $n$ $b$s, we have

$$h_n^1 = \underset{h \in \mathbb{R}^d}{\text{argmin}} \frac{1}{n+1} \left( |a_1 h_1 - a_{d+1}| + n|b_1 h_1 - b_{d+1}| \right), \tag{15}$$

where $h_1$ is the first element of $h$. We can rewrite the main part of the objective function as

$$|a_1 h_1 - a_{d+1}| + n|b_1 h_1 - b_{d+1}| = |a_1| \left| h_1 - \frac{a_{d+1}}{a_1} \right| + n|b_1| \left| h_1 - \frac{b_{d+1}}{b_1} \right|. \tag{16}$$

From this, one readily sees that the first coordinate of the minimizer $h_n^1$ equals $\frac{a_{d+1}}{a_1}$ if $|a_1| > n|b_1|$ and $\frac{b_{d+1}}{b_1}$ if $|a_1| < n|b_1|$. If $|a_1| = n|b_1|$, then it picks $\min(\frac{a_{d+1}}{a_1}, \frac{b_{d+1}}{b_1})$ as we are looking for the minimum-norm solution. For that same reason, all other entries of $h_n^1$ equal 0. Similar expressions, with $n-1$ substituted for $n$, hold for $h_{n-1}^1$. If we take $|b_1| < \frac{|a_1|}{n+1}$, then we get $(n+1)\ell(b, h_n^1) - n\ell(b, h_{n-1}^1) = (n+1)|\frac{a_{d+1}}{a_1} b_1 - b_{d+1}| - n|\frac{a_{d+1}}{a_1} b_1 - b_{d+1}| = |\frac{a_{d+1}}{a_1} b_1 - b_{d+1}|$, which is larger than 0 if $a_1 b_{d+1} \neq b_1 a_{d+1}$. Again along the same lines as for the squared loss, this shows that regression using the absolute loss is not locally $(\mathscr{Z}, \ell, n)$-monotonic and, as this holds for all $n$, we conclude that it is not weakly $(\mathscr{Z}, \ell, N)$-monotonic for any $N \in \mathbb{N}$.

Finally, the hinge loss. As we are necessarily dealing with a classification setting now, $a_{d+1}$ and $b_{d+1}$ are in $\{-1, +1\}$. Now, take $a_1 > 0$, $b_1 > 0$, $a_{d+1} = +1$ and $b_{d+1} = -1$. Any choice of $h$ can only classify either $a$ or $b$ correctly, as both $a_1$ and $b_1$ are positive. With this, the empirical risk becomes $\frac{1}{n+1} \left( \max(0, 1 - a_1 h) + n \max(0, 1 + b_1 h) \right)$ and only solutions $h$ for which the first coordinate is in $\left[ -\frac{1}{b_1}, \frac{1}{a_1} \right]$ need to be considered, as values outside of this interval will only increase the loss for either $a$ or $b$, while the loss remains the same for the other value. Being limited to the interval $\left[ -\frac{1}{b_1}, \frac{1}{a_1} \right]$ implies $\max(0, 1 - a_1 h) = 1 - a_1 h = |1 - a_1 h|$. So we will find exactly the same solutions as we found for the absolute loss, but with $a_{d+1}$ and $b_{d+1}$ limited to $\{-1, +1\}$. $\qquad\square$

**Theorem 3**

Take $a$ and $b$ to be in $B_0 \subset \mathscr{Z}$. As opposed to the proof for Theorem 2, we now cannot use the suggestion from Remark 2, as for the log-likelihood it does not hold that $\ell(b, h_n^0) = \ell(b, h_{n+1}^0) = 0$. Therefore, we need to look at the full expression of Lemma 1: $-\ell(b, h_{n+1}^0) + (n+1)\ell(b, h_n^1) - n\ell(b, h_{n-1}^1)$. The sigma that belongs to the empirical risk minimizing hypothesis $h_{n+1}^0$ equals $\sqrt{b^2}$.

For $h_{n-1}^1$ it is $\sqrt{\frac{a^2+(n-1)b^2}{n}}$ and for $h_n^1$ we get $\sqrt{\frac{a^2+nb^2}{n+1}}$. Therefore, we come to the following *negative* log-likelihoods:

$$\ell(b, h_{n+1}^0) = \log|b| + \frac{1}{2} + \frac{1}{2}(\log(2) + \log(\pi)), \tag{17}$$

$$\ell(b, h_n^1) = \frac{nb^2}{2\left(a^2 + (n-1)b^2\right)} + \log\left(\sqrt{\frac{a^2 + b^2(n-1)}{n}}\right) + \frac{1}{2}(\log(2) + \log(\pi)), \tag{18}$$

$$\ell(b, h_{n-1}^1) = \frac{(n+1)b^2}{2\left(a^2 + nb^2\right)} + \log\left(\sqrt{\frac{a^2 + b^2 n}{n+1}}\right) + \frac{1}{2}(\log(2) + \log(\pi)). \tag{19}$$

Now, consider the limit of $b$ going to 0. The last two negative log-likelihoods are finite in that case, while $\ell(b, h_{n+1}^0)$ will go to minus infinite. This implies that for $b > 0$ small enough, we have that $-\ell(b, h_{n+1}^0) + (n+1)\ell(b, h_n^1) - n\ell(b, h_{n-1}^1) > 0$ (because of the term $-\ell(b, h_{n+1}^0)$). In conclusion, our density estimator is not locally $(\mathscr{Z}, \ell, n)$-monotonic and, as this holds for all $n$, we conclude that it is not weakly $(\mathscr{Z}, \ell, N)$-monotonic for any $N \in \mathbb{N}$. □