[Reviews · NeurIPS 2019]

Reviewer 1



The paper considers the behaviour of learning curves, and formally defines a notion of risk monotonicity. It formally proves that for the commonly used square and hinge loss, these may be non-monotone for certain discrete distributions (i.e., that optimal performance may be achieved on a finite number of samples). Additional results are shown for density estimation, and several implications and resulting questions are discussed. The problem considered here is fundamental in nature, and concerns the intriguing phenomenon of performance getting worse with more samples. As noted in the introduction, this behaviour is non-intuitive, and certainly warrants careful study. The paper is rather well-written, and does a nice job of surveying the existing literature on this topic, some of which have demonstrated the existence of non-monotone learning curves. Compared to these works, the present paper aims to formally prove that non-monotonicity is possible for low-complexity learners, and for the loss being used for training (e.g., hinge). This is a reasonable setting for theoretical study, although distinct from the current trend in analysing generalisation properties of neural networks, such as the double-descent and jamming phenomena noted in line 99. While it would arguably be of more practical interest to directly study the latter, I think it of interest to demonstrate that non-intuitive behaviour is possible even in seemingly benign settings. The basic definition of monotonicity is sensible, and not particularly surprising: one asks that a learner yield lower expected risk (over the draw of the training sample) for any possible distribution over the inputs, and possibly for only sufficiently large sample sizes. Certainly it is useful to explicate the notion of monotonicity, but I did feel there was scope to shorten the discussion a bit to allow more space for the subsequent sections (see below). The theoretical results first present examples of learners that are provably risk-monotone, and then learners that are not. The former are useful to give some grounding as to where one obtains the expected behaviour, and include the simple case of estimating the mean of a Gaussian with known variance. It appeared that the discussion on fitting a categorical distribution was a bit more speculative: the authors state that "It therefore seems reasonable to expect that this learner acts risk monotonically. One route towards making a rigorous argument...". My suggestion is to compress or omit this case if a formal result is lacking, again to make space for the results that are actually proven, and also because the main interest in the paper is in the cases where we do not have monotonicity. The results for non-monotone learners involve a discrete distribution defined over two instances. Here, it is shown that unless the loss function satisfies a particular condition, one will not have monotonicity. At this stage, I found the discussion a little lacking: one has to wait until Sec 6 for the details of the result to get some mention, and even then the details of the setup considered remain mysterious. Specifically, it is not made particularly obvious how to interpret the condition on the loss in Lemma 1 or Remark 2, and to assess whether it is reflective of a scenario likely to be encountered in practice. As hinted in the discussion preceding, and explicated in the proof, the distribution where the bad behaviour is manifest involves there being two regions of space, one of which is very rare compared to the other. The condition on the loss is effectively a condition on how well the learner can perform on the dominant region, as a function of the sample size. The paper then uses this result to prove non-monotonicity is possible for learners using the square, hinge and absolute losses. Essentially, this relies on establishing that the condition on the loss function from Remark 2 is not satisfied for certain distributions. While the result appears correct, I again was hoping for some discussion on how to interpret the result: can one give some sense as to what is the quirk in the distribution which interacts badly with the studied losses? Though perhaps non-trivial to construct, it would've been ideal to see an example similar to that of [Loog and Duin 2012], Sec 2, which neatly provides intuition as to why non-monotonicity is possible for LDA. The empirical illustrations are on synthetic data, which I do not object to as means of demonstrating that non-intuitive phenomena are possible. Per earlier comments, one does wonder whether one can isolate specific properties of real-world distributions that increase the risk of non-monotone behaviour; and whether one can thus obtain similar results to the double-descent curve in the present setting. Minor comments: - line 15, "27" appears abruptly. - citation style, \citet is sometimes incorrectly used. - the paper notes that prior work on the possible non-monotonicity of curves relies on misspecified models. It was not clear if the the present paper fares differently. Is perhaps the point that the condition on the loss is an abstract statement that may hold whether or hold the Bayes-optimal predictor is in our class? - Sec 4.2 heading, append "monotonically" - "is a distribution where a very small part of the density is located relatively far away" -> it was not clear how exactly this is captured in Lemma 1. Presumably q -> 0 is meant to capture the "very small part of the density", but where is the assumption that the parts are "relatively far away"? Is this implicit in the assumption on the loss function? - Sec 4.2, for symmetry, consider starting with the case of fitting a Gaussian with unknown mean and variance, to act as counterpart to the beginning of Sec 4.1. - Sec 5, why do you need different values of q for the different losses? - a natural question is whether one can obtain stronger results about the shape of the learning curve (e.g., double-descent) in the simple scenarios considered here, rather than purely establishing that the behaviour is not monotone.

Reviewer 2



Update: I have reviewed the author feedback and was satisfied with some of the answers I have received. The motivation for the definition itself is still not clear enough to me (more specifically - its strictness) but I believe I can be patient and let the research of different but similar definitions be postponed to future work (as mentioned toward the end of the paper). Good luck. Original review below: The idea of the paper is interesting and discusses the fact that expected risk does not necessarily improve with larger data sets. It is clearly written and the points (and proofs) come across very neatly. On the other hand, the authors state that intuition would lead to the fact that the learning curve is monotonous. While it is an interesting discussion, I believe it is rather general to assume this and I believe that there are quite a few researchers that would put this statement to doubt. The analysis of the example distributions is interesting and clear, and the empirical results provide neat support to the proven theorems and examples. For this reason choose to accept this paper. The reason for my low acceptance score is that I believe the paper would have been more whole with more research of the exact conditions for monotonicity. The discussion that monotonicity is non-trivial is a large part of the paper which I believe could have been navigated towards the mentioned ideas. With further research regarding the conditions and the relation to PAC learnability (and combinatorial measures such as the VC dimension) I think this would be a very good paper. One last point which I would like to address: the notion of monotonicity seems rather strict. Would the proofs hold if instead of monotonicity you would require convergence of the risk? Best of luck

Reviewer 3



The authors introduced the formal notion of risk monotonicity, that the risk does not deteriorate with increasing training set sizes in expectation, which is a natural phenomena for learning curves. Then the authors presented a surprising result that various standard learners act non-monotonically. The authors provided a theoretical condition for non-monotonicicy with numerical experiments for classification, regression, and density estimation. Since the paper presents a natural framework of monotonicity and a strange counter-example of monotonicity, i.e. non-monotonic behavior of simple learners, originality and quality of the paper is quite high, and the paper is well-organized and its clarity and readability is sufficient. I think the notion of risk monotonicity includes a lot of important mathematical problems, however, significance for neurips community is unclear because presented numerical examples are too limited and not practical.

[Author Response · NeurIPS 2019]

We thank all three reviewers for their considerate appraisal and constructive comments. All agree that the idea presented in the paper is interesting. Reviewer #1 notes, in addition, that the work is fundamental in nature and that the discussed phenomenon is non-intuitive, intriguing, and warrants careful study. Reviewer #2 states that the analysis of the example distributions is interesting and clear. Reviewer #3 considers our results surprising and deems the originality and the quality of the work to be high. In what follows, we respond to the most important issues from the particular reviews.

**Reviewer #1**

We cannot do otherwise than agree with the sentiment expressed that more intuition into the when and why of (non)monotonicity in empirical risk minimization would be appreciated. This should also be apparent from the discussion that we provide in the paper. We do hope that our work spurs others to delve into what we think is an interesting topic. Indeed, the primary insight obtained in this submission is that nonmonotonicity at all happens, even in fairly standard settings. This is of course of interest in itself. At this point, we are simply unable, however, to offer any deeper intuition or to comment on the potential occurrence of nonmonotonicity in practice.

The reviewer's remark about identifying properties that increase the risk on nonmonotonicity is interesting. We will include it in the discussion as another direction for future research and relate it to our suggestion of $\mathcal{D}$-monotonicity, which aims to ensure monotonicity by making assumptions about the distributions considered. For further clarification, in advance of the discussion that comes with Equation (8) in Section 6, we will add one or two sentences directly after Lemma 1 and Remark 2. These will state that, to some extent, the results from the lemma and the remark show that if the learning of the single point $b$ does not happen fast enough, local monotonicity cannot be guaranteed. The necessary space for all additional text can be realized through the compression or even removal of (parts of) Section 3 and the categorical distribution example in Section 4 (as suggested by the reviewer).

We thank the reviewer for the minor comments provided, which we will use to further clarify our paper.

**Reviewer #2**

We agree that research into the exact conditions that allow for monotonicity would be most welcome. Similarly, it is of interest to further study the relationship between PAC-learnability and monotonicity. The discussion in our paper states some related directions of interest. Currently, however, we are simply not able to provide any further results into such directions. What we should emphasize more clearly in our paper though, is that PAC-learnability is an essentially different concept, independent of monotonicity. That is, learning problems of all four possible combinations exist: not PAC-learnable and monotonic, PAC-learnable and not monotonic, etc. For instance, the memorize algorithm (line 213 in our paper) is monotone, while it has infinite VC-dimension and so is not PAC-learnable. As such, combinatorial measures like the VC-dimension may not be useful in the analysis of monotonicity.

Incidentally, the reviewer could definitely be right in believing that there are quite a few researchers that would doubt that learning curves necessarily show improved performance with more data. We of course admit that our own believe is based on anecdotal evidence and personal experience—which should be clear from our submission as well. Indeed, up to now, we have met very few people who doubted the monotonicity statement and who were not surprised by our findings. Also, we do think the quote from the book by Shalev-Shwartz and Ben-David, both leading researchers in the field of learning theory, is telling. It states that the learning curve "must start decreasing once the training set size is larger than the VC-dimension" (see page 153 in the free online copy of the book; cited on page 1 of our paper).

Finally, we agree that monotonicity, especially its global form, is rather strict and it is of interest to think about relaxations. One option, making assumptions on the class of distributions, which we call $\mathcal{D}$-monotonicity, is already mentioned in the last section of our submission. The convergence of the risk (as suggested by the reviewer), however, seems a rather different notion that is already fulfilled by any learner that is consistent. We believe that our definition of local monotonicity is of interest exactly because it is not a limit property, but a finite sample size characteristic.

**Reviewer #3**

We could see that our work lacks some appeal for the practitioner and that it may be primarily the more theoretically inclined NeurIPS attendee that finds our results of interest. Still, our finding is for many so counterintuitive that it can even spark curiosity in the most ardent practitioner. Moreover, we believe that it is important that practitioners understand what type of learning curve behavior is possible, for which our work provides some first results.

Certainly, the two-point discrete distributions are merely constructs to show that nonmonotonic behavior can at all arise. They are definitely not practical. To throw light on more practical settings, we need to further characterize risk monotonicity. A full characterization of this phenomenon—something we also state in the paper's discussion—is of course an ultimate goal, since it would bring clarity to whether such behavior could occur in more realistic settings.

[Meta-Review · NeurIPS 2019]

This paper is a fascinating study that shows that the folk wisdom of risk monotonicity does not necessary always hold; more data can actually lead to worse performance for standard learners. This work is both technically solid and thought-provoking, making it a strong addition to the program. [This meta-review was reviewed and revised by the Program Chairs]